# Quality-of-Life Assessment in Children with Mild to Moderate Bronchial Asthma

**DOI:** 10.3390/medicina60050771

**Published:** 2024-05-07

**Authors:** Agnė Čibirkaitė, Vilija Bubnaitienė, Edita Hansted, Vaidotas Gurskis, Laimutė Vaidelienė

**Affiliations:** Department of Pediatrics, Medical Academy, Lithuanian University of Health Sciences, LT 44307 Kaunas, Lithuania; vilija.bubnaitiene@lsmu.lt (V.B.); edita.hansted@lsmu.lt (E.H.); vaidotas.gurskis@lsmu.lt (V.G.); laimute.vaideliene@lsmu.lt (L.V.)

**Keywords:** bronchial asthma, children, child, pediatric patient, quality of life

## Abstract

*Background and Objectives:* Assess the quality of life of children aged 2–10 with mild to moderate bronchial asthma. To evaluate the general health condition of children with mild and moderate severity bronchial asthma. To determine health changes in children with mild- and moderate-severity bronchial asthma as they grow older. To evaluate the impact of mild- and moderate-severity bronchial asthma on children’s daily and social activities, physical health, emotional state, and general well-being. *Materials and Methods*: A comparative cross-sectional study was conducted in March–June 2020. Parents or guardians of 2–10-year-old children without bronchial asthma and children with mild to moderate bronchial asthma were interviewed after receiving their written informed consent. The questionnaire was based on the standardized quality-of-life quiz SF-36. A total of 248 questionnaires were collected—106 from the parents or guardians of children with bronchial asthma and 142 from parents/guardians of children without bronchial asthma. For further analysis, 106 children without bronchial asthma and with no chronic conditions were selected. Quantitative variables were compared using the Mann–Whitney U test and qualitative data using the chi-square (χ^2^) criteria. Quantitative data were described by giving means, medians, and standard deviations (SD); qualitative features by giving relative frequencies. Statistical data were analyzed using SPSS and Excel 2020. *Results:* Children with mild and moderate asthma exhibit poorer health compared to their healthy counterparts. Only 20.7% of respondents with asthma reported excellent or very good health, contrasting with 64.1% of healthy children (*p* < 0.001). As children with asthma age, their general condition improves, with 46.2% showing improvement in the past year, while 42.5% of healthy children had a stable condition (*p* < 0.05). In various activities, children with asthma face more constraints than healthy children (*p* < 0.05), including energetic activities (sick—59.5%; healthy—10.3%), moderate activities (sick—24.5%; healthy—4.7%), climbing stairs (sick—22.7%; healthy—3.8%), and walking over 100 m (sick—9.4%; healthy—0%). Children with asthma are more likely to experience exhaustion, anxiety, tiredness, lack of energy, and restraint in public activities (*p* < 0.05). *Conclusions:* Parents/caregivers of children with mild to moderate bronchial asthma rate their health worse than those of healthy children do. As children with mild to moderate bronchial asthma grow, the disease impact on their overall well-being decreases. Children with mild to moderate bronchial asthma, compared to healthy children, experience more limitations in vigorous or moderate activities; face more difficulties climbing stairs or walking more than 100 m; frequently feel exhaustion, anxiety, fatigue, or lack of energy; and encounter restrictions in social activities.

## 1. Introduction

Bronchial asthma (BA) is a commonly occurring respiratory system disease, causing inflammation and narrowing of the airways [1,2,3]. According to the World Health Organization (WHO), in 2016, as many as 339 million people worldwide suffered from BA. In the same year, 417,918 people died from BA, with the majority living in less affluent countries [4]. Recently, with the increasing number of BA cases both in individual countries and worldwide, the concept of “asthma epidemic” has emerged [5,6,7].

The fact that the spread of asthma is rapidly increasing is also evidenced by the data from the Lithuanian Institute of Hygiene. In 2019, a total of 74,222 people in Lithuania were diagnosed with asthma or asthmatic condition. Meanwhile, in 2013, there were significantly fewer such individuals—64,726 people [8]. The mortality rate from BA in children in Lithuania is low—around 0.3 per 100,000 population [9].

The level of symptom control and quality of life of children with BA depends on accurate diagnosis and appropriately prescribed treatment [10,11]. Since there are still no BA diagnostic tests that meet the gold standard, cases of misdiagnosed BA occur. It is also difficult to assess respiratory system function in preschool-age children qualitatively. Therefore, it is necessary to know not only the basic principles of BA treatment but also the methods, criteria, and key aspects of diagnosis in children [12,13]. Assessing quality of life using standardized questionnaires is important in determining BA treatment and evaluating its effectiveness [14].

The essence of this work is to assess the quality of life of children aged 2–10 with mild to moderate BA: to evaluate general health condition in children with mild and moderate severity BA, determine health changes in children with mild- and moderate-severity BA as they grow older; and to evaluate the impact of mild- and moderate-severity BA on children’s daily and social activities, physical health, emotional state, and general well-being.

## 2. Materials and Methods

Study type and ethical considerations. A comparative cross-sectional study was conducted in Lithuania in March–June 2020. The Bioethics Centre of the Lithuanian University of Health Sciences permitted this research to be conducted on 21 January 2020, under permit number BEC-MF-235. Parents and guardians of 2–10-years-old children with or without BA were interviewed using electronic questionnaires. Before participating in the study, the parents/guardians of the participants read detailed information about the study and had to sign a written voluntary consent form to participate in the study. The participation rate was 90%. The main reasons for refusal were the length of the questionnaire and not having enough time.

Study setting. Survey questionnaires were electronic. All children with BA had visited a pediatric pulmonologist or allergist at least once in one of the major Lithuanian centers: Lithuanian University of Health Sciences (LUHS) Hospital Kaunas Clinics, Vilnius University Hospital Santaros Clinics, or Klaipeda Children’s Hospital. The International Classification of diseases Australian Modification (ICD-10-AM) code for asthma (J45) was entered into the electronic health system for all patients with asthma. The group of healthy children was under the care of their family doctors at their primary health-care institution. All parents and guardians of children with or without BA have been contacted and asked to participate in the study online.

Study instrument. The research instrument was a questionnaire consisting of four parts. The first part was designed to assess the general demographic characteristics of the subjects (gender, age). The second part was aimed at distinguishing the control group of healthy children from the group of children with mild to moderate BA. The next part was composed to evaluate the severity, symptoms, and control of BA in the affected individuals. The final, fourth part was based on the standardized quality-of-life questionnaire SF-36. This questionnaire, designed for use in clinical practice, is intended for both individual patients and scientific research aimed at assessing the benefits of various treatment methods or the burden of certain diseases on patients. The questionnaire assesses physical health, social functioning, full participation in various activities, overall health status, emotional well-being, and vitality (copyright belongs to The Medical Outcomes Trust (MOT), Health Assessment Lab (HAL) and Quality Metric Incorporated) [15]. In Lithuania the adaptation of the SF-36 questionnaire on linguistic and cultural bases was initiated at Vilnius University’s Institute of Experimental and Clinical Medicine in 2005. The questionnaire and answer options were translated from English into Lithuanian by two independent translators whose native language is Lithuanian [16]. Since this questionnaire is adapted for adults, during our study, the team of experienced pediatricians working at LUHS made some adjustments to certain questions to make them suitable for young children. For example, questions about physical activity were equated to active play and running, and questions about carrying objects were adjusted to carrying a backpack or heavier toys.

Study groups. In total, 248 respondents participated in the study. The questionnaire was completed by 106 parents/guardians of children with BA and 142 parents/guardians of children without BA. All children with BA did not have any concurrent chronic diseases. All members of the BA group met the criteria for mild to moderate BA:During the day, clinical BA symptoms occur daily or less frequently.During the night, clinical BA symptoms do not occur every night, but they occur every week or less frequently.BA symptoms are not provoked by light physical activity.Exacerbations are not frequent (<2 exacerbations in a past year that required hospitalization or oral corticosteroids).

Out of the 142 children without BA, 36 children were excluded from the further analysis because they had concurrent chronic endocrinological, pulmonological, neurological, or other illnesses. 

In the further analysis, two groups were distinguished:Children with BA: children aged 2–10 years with mild to moderate BA and without any other concurrent chronic diseases.Healthy children: children aged 2–10 years without BA and without any concurrent chronic diseases.

The average age of selected children with mild to moderate BA for statistical analysis was 5.9 years, while for healthy children, it was 4.5 years (see Figure 1).

## 3. Statistical Data Analysis

Statistical data analysis was performed using the SPSS statistical program (version number 26.0) and Microsoft Office Excel 2020. The samples (healthy children and children with mild to moderate BA) compared were large (*n* > 30) and independent. 

Quantitative variables were described by presenting the means, medians, and standard deviations (SD) of the compared samples. The normality of the age distribution was assessed using the Kolmogorov–Smirnov and Shapiro–Wilk tests, and showed significant results (*p* < 0.001 for both tests), indicating that the distribution of quantitative data was not normal. Consequently, to compare children with BA to healthy children, the Mann–Whitney U test was chosen, as it is a non-parametric test suitable for comparing two independent groups with non-normally distributed data.

For the distribution of sex, both normality tests also produced significant results (*p* < 0.001 for both tests), indicating non-normality. Due to that, the chi-square (χ^2^) test was deemed appropriate for comparing the proportions of sexes between the two groups, as it is robust against violations of normality assumptions.

In assessing the normality of the SF-36 questionnaire quality of life assessment results obtained from both groups, the Kolmogorov–Smirnov and Shapiro–Wilk tests were also conducted, revealing all *p*-values to be less than 0.001, indicating a departure from normality. Accordingly, to compare the qualitative data between children with mild- to moderate-severity BA and healthy children, the χ^2^ test was chosen. The equality of proportions was evaluated using the z-test with the Bonferroni method. Results were presented by providing the frequency and relative frequency of qualitative-feature values in the compared samples. Data were considered statistically significant when *p* < 0.05.

## 4. Results

General statistical data. In total, 106 patients with mild and moderate BA and 106 healthy children aged 2–10 were examined. The average age of BA group was 5.9 years (standard deviation (SD) = ±2.6; median = 6), while the average age of healthy children was 4.5 years (SD = ±1.9; median = 4). The minimum age for both groups of children was 2 years, and the maximum was 10 years. It has been determined that the age of sick and healthy children differs statistically significantly (*p* < 0.01). Due to that, it was essential to evaluate the impact of this on our results. Michel G. et al. [17] conducted a study comparing the quality of life of 21,590 children and adolescents living in 12 European countries, depending on their age and gender. No significant differences were found among children who had not yet reached adolescence. However, the conclusion was drawn that the quality of life of younger children is statistically significantly better (*p* < 0.001) than that of adolescents. Since adolescents did not participate in our study, we can state that the age difference of the subjects did not have a significant impact on the results.

Examining demographic characteristics revealed that in both the group of children with BA (*n* = 66, 62.3%) and the group of healthy children (*n* = 60, 56.6%), boys constituted the majority. In contrast, girls formed a smaller proportion (children with BA—*n* = 40, 37.7%; healthy children—*n* = 46, 43.4%). In both groups, the distribution by gender did not differ significantly (χ^2^ = 0.70; *p* = 0.401), indicating that gender had no influence on the obtained results. In a systematic review of the literature that assessed articles published from 1980 to 2007 that examined the characteristics of individuals with BA (gender, age), it was found that BA was more prevalent among boys until adolescence. After adolescence, girls were more likely to suffer from BA [18]. These review results align with our study data. We examined 2–10-year-old children with mild to moderate BA, where puberty had not yet begun. A larger proportion of children with BA were boys.

Assessment of bronchial asthma severity and treatment methods. To assess the severity and nature of symptoms experienced by individuals with BA during both the day and the night, as well as the characteristics of exacerbations, a survey was conducted among the group of individuals with BA. It aimed to determine the severity of the participants’ condition, which could potentially influence their assessment of their quality of life.

In evaluating the frequency of daytime BA symptoms (wheezing, shortness of breath, chest tightness, cough, etc.), most respondents indicated that clinically significant BA occurred less frequently than once a week (68%). Meanwhile, symptoms occurring more than once a week but less than once a day affected 14% of participants. Daily symptoms were experienced by 18%. Assessing clinical signs of nighttime BA symptoms revealed that the majority (74%) experienced symptoms rarely (less than twice a month), while 19% reported complaints occurring more than once every 2 months but less than once a week. Only a very small portion of participants (7%) experienced nighttime symptoms more than once a week.

Regarding BA exacerbations, most respondents indicated that exacerbations were short-term, of varying severity (37%), or could slightly limit a patient’s activity or negatively impact sleep (42%), while 21% mentioned that exacerbations not only affected physical activity and sleep but also led to a daily need for inhaled β^2^ agonists.

Overall, considering daily and nighttime symptoms as well as exacerbation characteristics, it can be stated that most individuals in the BA group suffered from mild episodic and mild persistent BA, with a smaller percentage experiencing moderate persistent BA.

Analyzing the treatment received by children with asthma, it was found that 15% of children (*n* = 16) used only Salbutamol inhalations on an as-needed basis when symptoms occurred. Additionally, 60% of patients (*n* = 64) took low doses (from 100 to 200 mcg per day) of inhaled glucocorticosteroids (ICS) daily, while 19% (*n* = 20) of children with asthma took medium doses (from 200 to 250 mcg per day) of ICS daily. Only a small fraction, 6% (*n* = 6), of respondents indicated that their children received not only low doses of ICS daily but also oral leukotriene receptor antagonists (LTRA) for additional treatment (Figure 2).

Evaluation of the quality of life in children with bronchial asthma, compared to healthy children. Analyzing the quality of life of children participating in the study, initial general questions were posed to determine how the parents/guardians assess their children’s health and how that assessment changes as the children grow. It was observed that among those with mild to moderate BA, eight respondents (7.5%) rated their children’s health as excellent, 14 (13.2%) as very good, 57 (53.8%) as good, 26 (24.5%) as not bad, and one (0.9%) as poor. Meanwhile, among the healthy group, a statistically significant number of respondents rated their children’s health as excellent or very good, compared to the asthmatic group—21 (19.8%) and 47 (44.3%) respondents, respectively. A smaller but statistically significant portion of the healthy group rated the child’s health as good or not bad—32 (30.2%) and six (5.7%), respectively. In the group of healthy children, none of the respondents rated rated their health poorly (Figure 3). A statistically significant difference was found between the examined groups (χ^2^ = 44.203, significance (*p*) < 0.001), indicating that the overall health assessment of children with mild and moderate BA was poorer than that of healthy children. 

When evaluating the changes in the health of the studied children as they grow, a statistically significant difference was also found (χ^2^ = 58.000, *p* < 0.05): among those with BA, more respondents (49, 46.2%) indicated that their children’s health was much better or slightly better than it was a year ago. In the healthy group, a smaller proportion of respondents (34, 32.1%) reported their health was much better than it was a year ago. Also, statistically significantly, the health of a larger portion of the control group of children remained stable over the year—45 (42.5%)—while only 20 respondents (18.9%) with BA indicated stability (Table 1). Therefore, it can be concluded that the health of children in the healthy group remained unchanged over the year, while in children with BA, health gradually improved as they grew (*p* = 0.003). During several different studies, it was found that the number of hospitalizations related to BA decreased among adolescents (13–18 years old) and young adults (19–30 years old) compared to young children [19,20]. This indicates that as children grow, BA symptoms tend to alleviate. This aligns with our study findings.

The study investigated the current overall health of children and its impact on various daily activities. It was found that children with BA were significantly (χ^2^ = 56.379, *p* < 0.001) more restricted in vigorous activities requiring endurance (e.g., running, active games, sports) compared to healthy children. Specifically, 10.4% of asthmatic children felt significantly restricted in energetic activities, while only 0.9% of healthy children felt similarly. Conversely, a statistically significant higher percentage of healthy children (89.6%) felt unrestricted compared to children with BA (40.6%). In moderate activities (e.g., cycling, pushing objects), it was also statistically significant (χ^2^ = 16.770, *p* < 0.001) that children with BA faced more difficulties in these daily activities: 23.6% of asthmatic children, compared to only 4.7% of healthy children, felt slightly restricted. A significantly larger portion of healthy children (95.3%, compared to 75.5% of those with BA) did not feel any restriction during moderate activities. Examining responses about the limitations that children experience, a significantly larger portion of those with BA found it challenging to climb stairs (among asthmatic children, 3.8% felt strongly restricted and 18.9% felt slightly restricted; among healthy children, 0% felt strongly restricted and 3.8% felt slightly restricted) (χ^2^ = 16.841, *p* < 0.001). When evaluating health difficulties during walking, it was observed that there was no significant difference between children with asthma and healthy children when walking a hundred meters. However, when walking several hundred meters or more than a kilometer, children with BA experienced more limitations: 9.4% (walking several hundred meters) and 32.1% (walking more than a kilometer) in the asthmatic group, respectively, felt slightly restricted, compared to 0% and 9.4% in the healthy group (*p* < 0.05). Analyzing limitations during bending/kneeling, bathing, and dressing, no statistically significant differences were found (*p* > 0.05) (Table 2). It can be concluded that children with BA, due to their overall health condition, faced more challenges than healthy children in both vigorous and moderate activities, climbing stairs, and walking more than a hundred meters. When comparing our results to those of other authors, we found a study that was conducted in Poland in 2009. The quality of life of 37 children with BA was compared to that of 100 healthy children. Similarly to our study, it was found that the physical fitness of healthy children was higher than that of children with BA (on average, healthy children spent 144 min per day engaged in physical activity, while those with BA spent only 81 min). However, no significant difference was observed when assessing activities that did not require significant physical effort, such as reading books, watching television, or playing computer games [21].

In evaluating the impact of overall children’s health on daily activities, the study examined the influence of physical health. It was found that 29.2% of children with BA sometimes experienced difficulties in physical activity, compared to only 6.6% of healthy children. Meanwhile, a significant 73.6% of healthy children never felt restricted, compared to 41.5% of those with BA (χ^2^ = 27.992, *p* < 0.001). However, when assessing the impact on play, the number of tasks performed, and the effort required in various activities, no significant differences were observed between the studied groups (*p* > 0.05) (Table 3).

When evaluating emotional health, it was found that 7.5% of children with BA often felt indecisive, whilst none of the healthy children felt that way (χ^2^ = 10.349, *p* < 0.05). No statistically significant difference was observed when assessing the impact of emotional health on children’s play and other activities, the number of tasks performed, diligence, social interaction, or feelings of fear and distance (*p* > 0.05) (Table 4). Therefore, it can be stated that BA increased indecisiveness in children, but it did not significantly affect their emotional well-being. On the contrary, Rene van Gent et al. conducted a study that compared healthy children aged 7–10 to children diagnosed or undiagnosed (but clinically symptomatic) with BA. It was found that healthy children evaluated their emotional well-being better than those experiencing BA symptoms did (*p* < 0.05) [22]. It is possible that differences in sample age could have influenced differences in the results between our study and the Rene van Gent study.

When examining the recent overall well-being of children, no significant differences in vitality were found between healthy and asthmatic children (*p* > 0.05). However, it was observed that a significant majority of healthy children (50.9%) almost never felt anxious, compared to those with BA (32.1%) (χ^2^ = 8.512, *p* < 0.05). In the past four weeks, 67.9% of healthy children and 54.7% of asthmatic children very often felt calm and peaceful. A significantly larger proportion of asthmatic children (21.7%, compared to 11.3% of healthy children) felt calm only occasionally (χ^2^ = 9.722, *p* < 0.05). More asthmatic children (20.8%) than healthy children (9.4%) felt energetic only occasionally (χ^2^ = 10.008, *p* < 0.05). Children with BA were more frequently exhausted and tired (4.7%, very often exhausted; 10.4%, tired) compared to healthy children (0%, very often exhausted; 0.9%, tired) (χ^2^_exhaustion_ = 11.083, *p*_exhaustion_ < 0.05, and χ^2^_fatigue_ = 9.920, *p*_fatigue_ < 0.05) (Table 5). Therefore, it can be stated that BA has a negative impact on a child’s overall well-being—the child experiences more anxiety, exhaustion, and fatigue, and less time feeling energetic.

When evaluating whether BA statistically significantly influences a child’s social activities (e.g., visiting friends or relatives) due to emotional or physical problems, it was found that a statistically significant larger proportion of healthy children (73.6% compared to 55.7% of those with BA) were never bothered by these issues in the past four weeks (χ^2^ = 9.218, *p* < 0.05) (Table 6). When comparing our results to other authors, similar results were found. In the previously mentioned study in Poland, as in our study, it was statistically significantly determined that healthy children spent more time per day (117 min) interacting with friends than children with BA did (35 min) [21].

## 5. Discussion

Our main goal during the study was to assess whether children with BA are more limited in various activities of daily life compared to healthy children, because BA is one of the most common chronic inflammatory lower respiratory tract diseases worldwide—about 4.3% (339 million) of people suffer from it [3,23]. About 4–5% of children with asthma experience severe symptoms [24]. Therefore, it is important to find ways to assess the activities in which children with the condition are restricted, to improve their quality of life as much as possible, periodically evaluating how it changes as the child grows and adjusting their specific asthma treatment accordingly.

The diagnosis of asthma is based on the presence of typical clinical symptoms and objective test results [25,26]. The most common symptoms of asthma are wheezing, prolonged expiration, shortness of breath, intolerance to physical exertion, temperature changes, chest tightness, and cough [3,11]. There is no gold-standard diagnostic test for asthma. One of the most important ways to assess respiratory function is spirometry. Performing the test in preschool-age (2–6 years) children is difficult because it is challenging for them to follow medical personnel instructions correctly. Therefore, asthma in this age group is mainly diagnosed according to the medical history of the disease and clinical examination findings [27].

Treatment for BA is stepped, considering the severity of BA. Short-acting inhaled β^2^ agonists (such as salbutamol) are prescribed at all treatment stages to quickly control BA symptoms. ICS are first-line drugs for all age groups in the treatment of mild, moderate, and severe BA. ICS improve symptoms control and reduce exercise-induced bronchoconstriction, the risk of asthma-related death, and the frequency and duration of hospitalizations. LTRA (such as montelukast) can be prescribed as a monotherapy (for mild persistent BA) or in combination with ICS (for moderate or severe BA treatment). Long-acting inhaled β^2^ agonists (LABA) (e.g., formoterol) can be prescribed, but monotherapy is not recommended due to insufficient effectiveness. LABA is used in combination with ICS for the treatment of moderate or severe BA. Monoclonal antibodies should be considered for children with severe allergic BA. It reduces the frequency of symptoms and the doses of medications used. Oral corticosteroids are prescribed for those with poorly controlled symptoms for a short period, under the supervision of a specialist doctor and after assessing the risk of side effects [28]. In our study, treatment for children with BA who participated was prescribed according to the mentioned recommendations, considering the symptoms identified during diagnosis. The slightly uncontrolled course of asthma could have a slight impact on quality of life. Therefore, BA treatment needs to be monitored more frequently, and transition to a higher level of treatment should be performed in time.

During our investigation, we tried to examine studies conducted in other countries and attempted to compare them with the results of our study. 

First, the improvement of health in BA group over the time was analyzed. In several different studies, it was found that the number of hospitalizations related to BA decreases among adolescents (13–18 years old) and young adults (19–30 years old) compared to young children [19,20]. This indicates that as children grow, BA symptoms tend to alleviate. This aligns with our study findings. The improvement in BA control over time can be explained by both anatomical changes in the respiratory tract as the child grows and better medical control of the disease by applying the most suitable individualized treatment.

We also compared our results regarding subjects’ physical health and social relations. In a study conducted in Poland in 2009, the quality of life of 37 children with BA was compared to that of 100 healthy children. As in our study, it was found that the physical fitness of healthy children was higher than that of children with BA (on average, healthy children spent 144 min per day engaged in physical activity, while those with BA spent only 81 min). However, no significant difference was observed when assessing activities that did not require significant physical effort, such as reading books, watching television, or playing computer games. Also, in this study, as in our study, it was statistically significantly determined that healthy children spent more time per day (117 min) interacting with friends than children with BA (35 min) [21]. It is essential to consider the limitations of this study: the severity of BA was not specified, and the sample of participants with BA was not very large.

While investigating our findings on emotional well-being, we analyzed a study of Rene van Gent et al. They compared healthy children aged 7–10 to children diagnosed or undiagnosed (but clinically symptomatic) with BA. It was found that healthy children evaluated their emotional well-being better than those experiencing BA symptoms did (*p* < 0.05) [22]. On the contrary, our study has shown that BA increases indecisiveness in children, but it does not significantly affect their emotional well-being. To identify the possible causes of differences in the results, we examined another article that described a study conducted in 2015. It investigated the quality of life of 192 children aged 5–12 with BA. It was found that older children rated their quality of life worse than younger ones. The authors emphasized that, when evaluating not only preschool and early-school-age children but also older school-age children, it is necessary to consider factors affecting the quality of life, such as emotional state, changes in feelings as they grow, and transition to adolescence [29]. Since our study analyzed children aged 2–10, and the study by Rene van Gent examined an older sample (children aged 7–10), it is possible that differences in sample age could have influenced differences in the results between the two studies.

The data obtained in our study align with the conclusions of other authors that as children grow, BA symptoms alleviate; the physical fitness of healthy children is higher than those with BA; and individuals with BA are more restricted in social activities. 

To assess the consequences of the disease and the effectiveness of treatment, objective clinical research, various laboratory, and instrumental tests have been used to this day. These are integral and significant components of modern medicine. Although the mentioned methods greatly facilitate assistance to the patient, health care professionals often forget that the quality of life, and how it changes for people when they become ill or during treatment, is important for both prescribing treatment and evaluating its effectiveness [14]. Even as the patients recover and their test results improve, the damage to emotional health caused by the disease, feelings of isolation, and financial burden on the family, especially for a child, can significantly impact their well-being. Evaluating the quality of life can help identify the challenges patients face and provide the necessary comprehensive assistance [30].

In our study, we attempted to adapt a standardized quality-of-life assessment questionnaire SF-36 for children to enable the most objective evaluation possible of the difficulties in life caused by BA. Asthma is a chronic disease and BA treatment can last for many years, often for a lifetime, so it is important to take measures to enable patients to participate as much as possible in societal activities and to minimize the impact of the disease on their quality of life. Consequently, it is crucial to ensure regular visits to specialized doctors for those affected, involving a multidisciplinary team in the treatment, including pediatric pulmonologists or pediatric allergists, social pediatricians, psychologists, and family doctors.

## 6. Practical Application

It is important to find ways to assess the activities in which children with BA are restricted, to improve their quality of life as much as possible, periodically evaluating how it changes as the child grows.

Health care professionals often forget that the quality of life, and how it changes for people when they become ill or during treatment, is important for both prescribing treatment and evaluating its effectiveness. Evaluating the quality of life can help identify the challenges patients face and provide the necessary comprehensive assistance.

BA treatment can last for many years, often for a lifetime, so it is important to take measures to enable patients to participate as much as possible in societal activities and to minimize the impact of the disease on their quality of life. Consequently, it is crucial to ensure regular visits to specialized doctors for those affected, as BA treatment needs to be monitored frequently, and transition to a higher level of treatment should be done in time.

## 7. Conclusions

Parents/caregivers of children with mild to moderate bronchial asthma rate their health worse than those of healthy children do.

As children with mild to moderate bronchial asthma grow, the disease impact on their overall well-being decreases.

Children with mild to moderate bronchial asthma, compared to healthy children, experience more limitations in vigorous or moderate activities; face more difficulties climbing stairs or walking more than 100 m; frequently feel exhaustion, anxiety, fatigue, or lack of energy; and encounter restrictions in social activities.

Our research has shown that bronchial asthma can significantly restrict a child’s life in various areas. Therefore, it is crucial to ensure regular visits to specialized doctors for those affected, involving a multidisciplinary team in the treatment, including pediatric pulmonologists or pediatric allergists, social pediatricians, psychologists, and family doctors.

### Limitations of the Study

The questionnaires were filled out by the parents/caregivers, not the subjects themselves, so the assessment of the child’s quality of life may be subjective and dependent on the opinion of parent/caregiver. Addressing this limitation with young children is challenging because they cannot fill out questionnaires themselves. To make the study as objective as possible, a standardized quality-of-life assessment questionnaire (SF-36) was used.

The average age between the compared groups differed statistically significantly, but the impact was minimal since the children who were studied had not yet reached adolescence.

Most children with BA were younger than 6 years old (average age, 5.9 years); due to that, the diagnosis of BA was established based on clinical and objective examination results, as spirometry is not performed in young children due to the specificity of the examination and the need for the patient to follow instructions. To refine the group of children with BA, older children should be examined, but then the results of the study would be influenced by adolescence.

The control group of healthy children was formed based on the absence of any chronic diseases’ diagnoses in the electronic health system and the parents’ description of the child’s health. To further refine the healthy children group, the study results would be made more reliable by selecting children who had physically visited a pediatrician at the LUHS Hospital Kaunas Clinics.

## Figures and Tables

**Figure 1 medicina-60-00771-f001:**
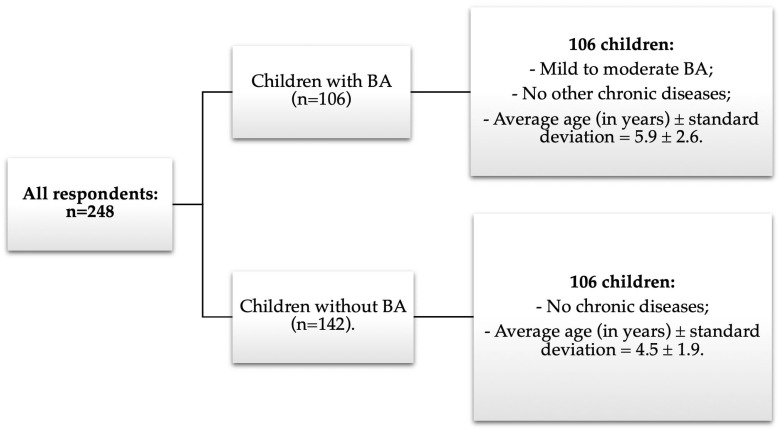
Schematic representation of sample distribution into groups.

**Figure 2 medicina-60-00771-f002:**
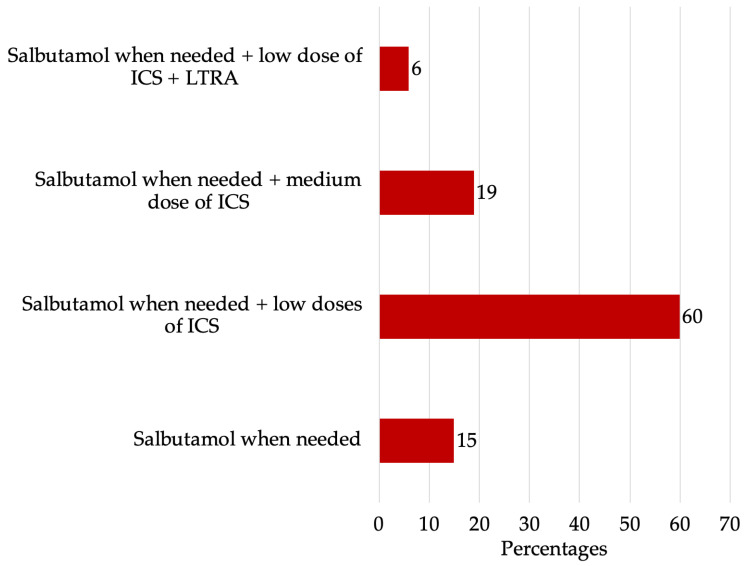
The characteristics of BA treatment methods in percentages.

**Figure 3 medicina-60-00771-f003:**
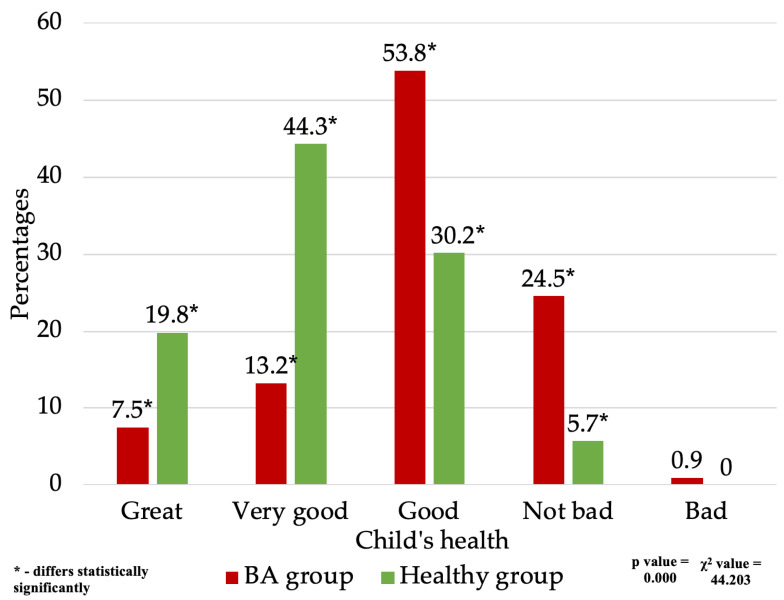
Child’s health assessment.

**Table 1 medicina-60-00771-t001:** Current child’s health compared to health a year ago.

Comparison of Child Health over the Period of One Year	Distribution of Respondents (in Percentages)		*p*-Value; χ^2^ Value
	BA Group	Healthy Group	
Now much better than a year ago	*n* = 49; 46.2% *	*n* = 34; 32.1% *	0.003; 58.000
Now slightly better than a year ago	*n* = 28; 26.4%	*n* = 19; 17.9%	
About the same as a year ago	*n* = 20; 18.9% *	*n* = 45; 42.5% *	
Now slightly worse than a year ago	*n* = 9; 8.5%	*n* = 8; 7.5%	
Now much worse than a year ago	*n* = 0; 0%	*n* = 0; 0%	

*—differs statistically significantly.

**Table 2 medicina-60-00771-t002:** Influence of child’s health on various activities.

Field of Activity	Distribution of Respondents (in Percentages)						*p*-Value; χ^2^ Value
	BA Group			Healthy Group			
	Severely Restricted	Slightly Restricted	Unrestricted	Severely Restricted	Slightly Restricted	Unrestricted	
Energetic activity	*n* = 11; 10.4% *	*n* = 52; 49.1% *	*n* = 43; 40.6% *	*n* = 1; 0.9% *	*n* = 10; 9.4% *	*n* = 95; 89.6% *	0.000; 56.379
Moderate activity	*n* = 1; 0.9%	*n* = 25; 23.6% *	*n* = 80; 75.5% *	*n* = 0; 0%	*n* = 5; 4.7% *	*n* = 101; 95.3% *	0.000; 16.770
Staircase climbing	*n* = 4; 3.8% *	*n* = 20; 18.9% *	*n* = 82; 77.4% *	*n* = 0; 0% *	*n* = 4; 3.8% *	*n* = 102; 96.2% *	0.000; 16.841
Bending and squatting	*n* = 1; 0.9%	*n* = 4; 3.8%	*n* = 101; 95.3%	*n* = 0; 0%	*n* = 2; 1.9%	*n* = 104; 98.1%	0.425; 1.711
Walking more than a kilometer	*n* = 7; 6.6% *	*n* = 34; 32.1*%	*n* = 65; 61.3% *	*n* = 0; 0% *	*n* = 10; 9.4% *	*n* = 96; 90.6% *	0.000; 26.060
Walking several hundred meters	*n* = 0; 0%	*n* = 10; 9.4%*	*n* = 96; 90.6% *	*n* = 0; 0%	*n* = 0; 0% *	*n* = 106; 100% *	0.001; 10.495
Walking one hundred meters	*n* = 1; 0.9%	*n* = 2; 1.9%	*n* = 103; 97.2%	*n* = 0; 0%	*n* = 0; 0%	*n* = 106; 100%	0.218; 3.043
Bathing or dressing	*n* = 1; 0.9%	*n* = 7; 6.6%	*n* = 98; 92.5%	*n* = 0; 0%	*n* = 3; 2.8%	*n* = 103; 97.2%	0.256; 2.724

*—differs statistically significantly.

**Table 3 medicina-60-00771-t003:** Impact of a child’s physical health on daily activities in the last 4 weeks.

Impact on Daily Activities		Distribution of Respondents (in Percentages)		*p*-Value; χ^2^ Value
		BA Group	Healthy Group	
The child spent less time on play or other activities	All the time	*n* = 0; 0%	*n* = 0; 0%	0.144; 5.408
	Very often	*n* = 3; 2.8%	*n* = 1; 0.9%	
	Sometimes	*n* = 28; 26.4%	*n* = 18; 17.0%	
	Almost never	*n* = 28; 26.4%	*n* = 25; 23.6%	
	Never	*n* = 47; 44.3%	*n* = 62; 58.5%	
The child performed fewer tasks/activities than desired	All the time	*n* = 0; 0%	*n* = 0; 0%	0.054; 7.647
	Very often	*n* = 5; 4.7%	*n* = 0; 0%	
	Sometimes	*n* = 24; 22.6%	*n* = 16; 15.1%	
	Almost never	*n* = 28; 26.4%	*n* = 34; 32.1%	
	Never	*n* = 49; 46.2%	*n* = 56; 52.8%	
You noticed that the child felt restricted (due to physical health) in a certain job or other activity	All the time	*n* = 0; 0%	*n* = 0; 0%	0.000; 27.992
	Very often	*n* = 7; 6.6%	*n* = 2; 1.9%	
	Sometimes	*n* = 31; 29.2% *	*n* = 7; 6.6% *	
	Almost never	*n* = 24; 22.6%	*n* = 19; 17.9%	
	Never	*n* = 44; 41.5% *	*n* = 78; 73.6% *	
The child encountered difficulties or had to put in more effort in a certain activity/task/game	All the time	*n* = 0; 0%	*n* = 1; 0.9%	0.216; 5.781
	Very often	*n* = 4; 3.8%	*n* = 2; 1.9%	
	Sometimes	*n* = 26; 24.5%	*n* = 19; 17.9%	
	Almost never	*n* = 29; 27.4%	*n* = 22; 20.8%	
	Never	*n* = 47; 44.3%	*n* = 62; 58.5%	

*—differs statistically significantly.

**Table 4 medicina-60-00771-t004:** Impact of the child’s emotional state on daily activities in the last 4 weeks.

Emotional State		Distribution of Respondents (in Percentages)		*p*-Value; χ^2^ Value
		BA Group	Healthy Group	
The child spent less time on play or other activities	All the time	*n* = 0; 0%	*n* = 0; 0%	0.127; 5.698
	Very often	*n* = 7; 6.6%	*n* = 2; 1.9%	
	Sometimes	*n* = 30; 28.3%	*n* = 21; 19.8%	
	Almost never	*n* = 23; 21.7%	*n* = 29; 27.4%	
	Never	*n* = 46; 43.4%	*n* = 54; 50.9%	
The child performed fewer tasks/activities than desired	All the time	*n* = 1; 0.9%	*n* = 0; 0%	0.560; 2.989
	Very often	*n* = 6; 5.7%	*n* = 3; 2.8%	
	Sometimes	*n* = 28; 26.4%	*n* = 24; 22.6%	
	Almost never	*n* = 28; 26.4%	*n* = 28; 26.4%	
	Never	*n* = 43; 40.6%	*n* = 51; 48.1%	
The child performed a task or engaged in another activity less diligently than usual	All the time	*n* = 0; 0%	*n* = 0; 0%	0.406; 2.907
	Very often	*n* = 7; 6.6%	*n* = 2; 1.9%	
	Sometimes	*n* = 29; 27.4%	*n* = 31; 29.2%	
	Almost never	*n* = 24; 22.6%	*n* = 25; 23.6%	
	Never	*n* = 46; 43.4%	*n* = 48; 45.3%	
The child was disturbed or withdrawn	All the time	*n* = 0; 0%	*n* = 0; 0%	0.416; 2.845
	Very often	*n* = 8; 7.5%	*n* = 4; 3.8%	
	Sometimes	*n* = 37; 34.9%	*n* = 46; 43.4%	
	Almost never	*n* = 23; 21.7%	*n* = 24; 22.6%	
	Never	*n* = 38; 35.8%	*n* = 32; 30.2%	
The child was scared	All the time	*n* = 0; 0%	*n* = 0; 0%	0.460; 2.586
	Very often	*n* = 6; 5.7%	*n* = 2; 1.9%	
	Sometimes	*n* = 28; 26.4%	*n* = 33; 31.1%	
	Almost never	*n* = 31; 29.2%	*n* = 33; 31.1%	
	Never	*n* = 41; 38.7%	*n* = 38; 35.8%	
The child felt different from others or isolated	All the time	*n* = 2; 1.9%	*n* = 0; 0%	0.138; 6.957
	Very often	*n* = 3; 2.8%	*n* = 0; 0%	
	Sometimes	*n* = 12; 11.3%	*n* = 7; 6.6%	
	Almost never	*n* = 16; 15.1%	*n* = 16; 15.1%	
	Never	*n* = 73; 68.9%	*n* = 83; 78.3%	
The child felt indecisive	All the time	*n* = 0; 0%	*n* = 0; 0%	0.016; 10.349
	Very often	*n* = 8; 7.5% *	*n* = 0; 0% *	
	Sometimes	*n* = 19; 17.9%	*n* = 27; 25.5%	
	Almost never	*n* = 14; 13.2%	*n* = 19; 17.9%	
	Never	*n* = 65; 61.3%	*n* = 60; 56.6%	

*—differs statistically significantly.

**Table 5 medicina-60-00771-t005:** Child’s overall well-being in the last 4 weeks.

Child’s Overall Well-Being		Distribution of Respondents (in Percentages)		*p*-Value; χ^2^ Value
		BA Group	Healthy Group	
The child felt energetic	All the time	*n* = 42; 39.6%	*n* = 52; 49.1%	0.152; 6.702
	Very often	*n* = 53; 50.0%	*n* = 51; 48.1%	
	Sometimes	*n* = 8; 7.5%	*n* = 2; 1.9%	
	Almost never	*n* = 2; 1.9%	*n* = 0; 0%	
	Never	*n* = 1; 0.9%	*n* = 1; 0.9%	
The child was very anxious	All the time	*n* = 0; 0%	*n* = 0; 0%	0.037; 8.512
	Very often	*n* = 7; 6.6%	*n* = 3; 2.8%	
	Sometimes	*n* = 31; 29.2%	*n* = 25; 23.6%	
	Almost never	*n* = 34; 32.1% *	*n* = 54; 50.9% *	
	Never	*n* = 34; 32.1%	*n* = 24; 22.6%	
The child felt calm and peaceful	All the time	*n* = 16; 15.1%	*n* = 19; 17.9%	0.045; 9.722
	Very often	*n* = 58; 54.7% *	*n* = 72; 67.9% *	
	Sometimes	*n* = 23; 21.7% *	*n* = 12; 11.3% *	
	Almost never	*n* = 5; 4.7%	*n* = 3; 2.8%	
	Never	*n* = 4; 3.8% *	*n* = 0; 0% *	
The child was very energetic	All the time	*n* = 27; 25.5%	*n* = 34; 32.1%	0.040; 10.008
	Very often	*n* = 53; 50.0%	*n* = 62; 58.5%	
	Sometimes	*n* = 22; 20.8% *	*n* = 10; 9.4% *	
	Almost never	*n* = 3; 2.8%	*n* = 0; 0%	
	Never	*n* = 1; 0.9%	*n* = 0; 0%	
The child felt exhausted	All the time	*n* = 0; 0%	*n* = 0; 0%	0.019; 9.920
	Very often	*n* = 5; 4.7% *	*n* = 0; 0% *	
	Sometimes	*n* = 28; 26.4%	*n* = 17; 16.0%	
	Almost never	*n* = 29; 27.4%	*n* = 41; 38.7%	
	Never	*n* = 44; 41.5%	*n* = 48; 45.3%	
The child felt tired	All the time	*n* = 0; 0%	*n* = 0; 0%	0.011; 11.083
	Very often	*n* = 11; 10.4% *	*n* = 1; 0.9% *	
	Sometimes	*n* = 51; 48.1%	*n* = 56; 52.8%	
	Almost never	*n* = 30; 28.3%	*n* = 40; 37.7%	
	Never	*n* = 14; 13.2%	*n* = 9; 8.5%	

*—differs statistically significantly.

**Table 6 medicina-60-00771-t006:** Negative impact of physical or emotional issues on social activities in the last 4 weeks.

Negative Impact of Physical or Emotional Issues on Social Activities	Distribution of Respondents (in Percentages)		*p*-Value; χ^2^ Value
	BA Group	Healthy Group	
Persistently posed challenges	*n* = 0; 0%	*n* = 0; 0%	0.027; 9.218
Very frequently posed challenges	*n* = 3; 2.8%	*n* = 0; 0%	
Occasionally posed challenges	*n* = 15; 14.2%	*n* = 9; 8.5%	
Almost never posed challenges	*n* = 29; 27.4%	*n* = 19; 17.9%	
Never posed challenges	*n* = 59; 55.7% *	*n* = 78; 73.6% *	

*—differs statistically significantly.

## Data Availability

The original contributions presented in the study are included in the article, further inquiries can be directed to the corresponding author.

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
