# Peer review of "Quality-of-Life Assessment in Children with Mild to Moderate Bronchial Asthma"

_medicina, 2024, doi:10.3390/medicina60050771_

Round 1

Reviewer 1 Report

Comments and Suggestions for Authors

Dear Editors, respected authors,

The work (ID: medicina-2971141) in this form, which was submitted for review, cannot be reviewed, since the work lacks the Introduction section (with the objectives of the work).

Some remarks: 

  • Return the paper for compliance with the Instructions to authors for writing papers for the journal.
  • Authors must pay attention that the Methods section lacks basic information about the study setting, study design, study population, study sample, ..., inclusion criteria and exclusion criteria from the study, description of the questionnaire that was applied in this study (include information about validation of the questionnaire, with appropriate references), with details on the recruitment of participants in the study, ethical considerations (whether and how obtained voluntary, informed, written consent to participate in the study).
  • References in this paper: the first reference in this paper is cited on page 12 in the Discussion section, only 7 references are cited in the paper in total, it is important that the references must be recent.
  • Pay special attention to Limitations of this study (list them, discuss them, list possibilities for overcoming them).  
  • In general, authors must follow the Instructions to authors for writing a paper and revise the paper in detail, so that the review of the paper can be conducted in a correct and high-quality manner.

Author Response

The following revisions have been made in the article:

  1. The introduction section has been written based on the latest literature sources, reflecting the research aims and objectives.
  2. The methods section has been improved:
    • the information about study setting, design, samples, inclusion and exclusion criteria has been given;
    • the research questionnaire has been described, information about the standardized questionnaire used in the study has been given and supported by literature.
  3. The article now states that informed consent was obtained from the parents/caregivers of the subjects.
  4. The list of literature used has been expanded (15 references in total).
  5. The limitations of the study have been considered, and measures have been implemented to minimize their impact.

Reviewer 2 Report

Comments and Suggestions for Authors

Dear Authors,

the manuscript does not have any introduction describing the background of the study, hypothesis generation and the declaration of the first aim of the study. This would prevent the possibility to understand and evaluate the appropriateness of the methods and then the obtained results. Please provide an introduction with a clear definition of the aim of the study.

In addition, please note that methods must be supported by the literature. Please refer to and quote published papers.

Author Response

The following revisions have been made in the article:

  1. The introduction section has been written based on the latest literature sources, reflecting the research aims and objectives.
  2. The methods section has been improved:
    • the information about study setting, design, samples, inclusion and exclusion criteria has been given;
    • the research questionnaire has been described, information about the standardized questionnaire used in the study has been given and supported by literature.

Round 2

Reviewer 1 Report

Comments and Suggestions for Authors

The authors of the paper (ID: medicina-2971141) must carefully prepare their work for consideration for publication in an international journal.

In a previous review that referred to a version of the paper that was not complete, several comments were made to the authors. 

However, the paper still lacks key information on the following issues: 

1. Study setting (where was this work carried out, in which health institution were the patients with asthma recruited, and in which institution/health institution were the healthy children recruited - the control group?); 

2. Ethical considerations: in the revised version of this paper, information is missing that a written, voluntary consent to participate in this study was obtained. It was only stated that informed consent was provided. Authors must provide detailed information on whether they had voluntary WRITTEN consent to participate in the study. All the circumstances on this matter must be stated in detail in the paper. 

3. State the `Participation rate` and explain whether and for what reasons there was a refusal to participate in this study; 

4. It is not stated whether the SF-36 questionnaire that was applied in this paper was previously validated in the Lithuanian language, as well as for the population in this study. Provide a corresponding reference. Why is the inappropriate reference number 16 (in English) given?  

5. The Discussion section must be reconstructed, and the presented results must be commented on in a logical order. 

Check all references. 

For example: on Line 264 it is stated `Michel G. et al. [1] conducted a study comparing the quality of life of 21,590 children`, is that reference number 1 from the list of references?  

Further, for example: does the text on Lines 285-293, for Poland, correspond to reference number 5 in the list of references? 

Further, for example: whether for the text on Lines 294-297, cited authors Rene van Gent et al. match reference number 6 in the reference list? 

Author Response

The following corrections were made:

  1. The section on research methods was improved: describing in what health care institutions patients were tracked and how two samples were refined.
  2. Ethical considerations were provided, emphasizing that informed written consent was obtained from parents/guardians.
  3. The participation rate was indicated.
  4. The SF-36 questionnaire was described in more detail, including the translation into Lithuanian details and the description of its’ adaptation for children.
  5. The discussion section was improved, reviewing the use of literature sources, verifying the literature list to match the cited sources.

The corrections to the paper were highlighted in yellow for the convenience of the reviewers.

Reviewer 2 Report

Comments and Suggestions for Authors

The manuscript is improved. However there are some more revisions required.

1) Please define in methods (inclusion criteria of group 1) how mild to moderate BA is defined.

2) please note that once you use an abbreviation, it must be used throughout the entire manuscript. For example, Bronchial Asthma (BA) must be corrected in the methods (inclusion criteria).

3) data must be reported as parametric (mean and SD) or non-parametric (median and interquartile range), based on a normality test. Please review the statistical method (add the assessment of normality) and apply the appropriate data expression

4) in statistical methods the authors state: "Quantitative variables were de-100 scribed by presenting the means, minimum and maximum values, and standard devia-101 tions (SD) of the compared samples. ". However this is not followed in the manuscript (for example: "The average age of patients was 5.9 years 111 (standard deviation (SD) = ± 2.6; median = 6), while the average age of healthy children 112 was 4.5 years (SD = ± 1.9; median = 4). "). Check the manuscript for its clarity

5) Figure 2 and 3: I think that "time/week" would mean "episodes/week"... check and correct please.

6) Figure 2, 3 and 4 could be easily included in the manuscript. I do not believe they are necessary for the manuscript.

7) Table 1 would be better to be reported as figure (this would improve the clarity and it would better focus the attention of the reader to your first studty aim)

8) Authors should describe the medical treatments of BA patients in a dedicated Table/text. This is important in the description of recruited population. If possible, add also some test/examination to better describe the population. The reader must clearly understand the characteristics of the population of the study.

9) Discussion should start with a short paragraph that summarizes the main findings of the study (reflecting the first and secondary aims)

Author Response

The following corrections were made:

  1. We clarified the severity criteria for BA in the methods section.
  2. "BA" abbreviation is used throughout the article.
  3. Data normality was checked using Kolmogorov-Smirnov and Shapiro-Wilk tests, and appropriate methods were selected for further data analysis.
  4. Quantitative indicator description parameters were adjusted.
  5. Figures 2, 3, and 4 were removed from the article.
  6. Table 1 was converted into Figure.
  7. Treatment methods applied to BA group patients were described and presented in the figure.
  8. The discussion section was improved, reviewing the use of literature sources, verifying the literature list to match the cited sources.

The corrections to the paper were highlighted in yellow for the convenience of the reviewers.